# Dual Zero-Watermarking Scheme for Two-Dimensional Vector Map Based on Delaunay Triangle Mesh and Singular Value Decomposition

**Xu Xi [1], Xinchang Zhang [2,3,*], Weidong Liang [1], Qinchuan Xin [1,*] and Pengcheng Zhang [4]**

[1] School of Geography and Planning, Sun Yat-sen University, Guangzhou 510275, China; xixu2016sysu@outlook.com (X.X.); xixu2013@163.com (W.L.)
[2] School of Geographical Sciences, Guangzhou University, Guangzhou 510006, China
[3] The College of Environment and Planning of Henan University, Henan University, Kaifeng 475000, China
[4] Guangzhou Urban Planning and Design Survey Research Institute, Guangzhou 510060, China; guangzhou2000@126.com
[*] Correspondence: eeszxc2018sysu@outlook.com (X.Z.); xinqinchuan@gmail.com (Q.X.); Tel.: +86-208-411-5103 (X.Z.)

**Abstract:** Digital watermarking is important for the copyright protection of electronic data, but embedding watermarks into vector maps could easily lead to changes in map precision. Zero-watermarking, a method that does not embed watermarks into maps, could avoid altering vector maps but often lack of robustness. This study proposes a dual zero-watermarking scheme that improves watermark robustness for two-dimensional (2D) vector maps. The proposed scheme first extracts the feature vertices and non-feature vertices of the vector map with the Douglas-Peucker algorithm and subsequently constructs the Delaunay Triangulation Mesh (DTM) to form a topological feature sequence of feature vertices as well as the Singular Value Decomposition (SVD) matrix to form intrinsic feature sequence of non-feature vertices. Next, zero-watermarks are obtained by executing exclusive disjunction (XOR) with the encrypted watermark image under the Arnold scramble algorithm. The experimental results show that the scheme that synthesizes both the feature and non-feature information improves the watermark capacity. Making use of complementary information between feature and non-feature vertices considerably improves the overall robustness of the watermarking scheme. The proposed dual zero-watermarking scheme combines the advantages of individual watermarking schemes and is robust against such attacks as geometric attacks, vertex attacks and object attacks.

**Keywords:** digital data protection; watermark; vector map; delaunay triangulation mesh; singular value decomposition

## 1. Introduction

Two-dimensional (2D) vector maps have a broad range of applications, such as engineering construction, urban planning, military affairs, and navigation [1–3]. The production of vector maps is often expensive, and illegal copying, altering, and misuse of the data could result in tremendous losses to data owners. With increasing demands for copyright protection, digital watermarking techniques have been developed rapidly in recent years [4–7]. Efforts have been made to design various watermark embedding algorithms for digital images to improve the robustness and imperceptibility of the watermark [8–12]. The abundant studies on image watermarking provide valuable information and references for other digital products as well [13–16], and thus enhance the development of watermarking for vector maps. Compared with watermarking of digital images, watermarking of

vector maps requires not only external graphic analysis but also internal numerical calculations and thus poses challenges to the watermarking schemes.

Classic watermarking algorithms of vector data mainly include the coordinate-domain and the frequency-domain watermarking algorithms. Cox and Jager [17] first proposed the coordinate-domain watermarking algorithm of vector maps and advocated encoding watermark information directly at each vertex coordinate. To improve the robustness of watermarking against various attacks, later studies developed methods to embed watermarks by controlling the position, tolerance, and accuracy of the map elements [18–20]. However, there is a lack of discussion on the deformation of map elements. For example, the Least Significant Bit method, a common watermarking algorithm in vector maps [6,21,22], embeds a watermark in the coordinate precision bit, but it is vulnerable to the bit-erase attack. In recent years, the developments in coordinate-domain watermarking algorithms have accounted for geographical features, geometric shapes and topological relations of map elements. Shao et al. [23] proposed a watermarking scheme for the vector map that modifies the statistics of non-feature vertices in data blocks while keeping the geometric shapes of geographical elements. Ohbuchi et al. [24] applied a general rule of the actual geometric error (0.75 m) in the 1: 2500 scale maps to limit the vertex movements after watermarking. Lee and Kwon [25] put forward a watermarking scheme for polylines and polygons using metrics of both relative and absolute position accuracies to ensure the vertex movements within a maximum tolerance range. Huber et al. [26] used the Voronoi diagram and constraint Delaunay triangulation to generate a maximum perturbed region for each individual vertex, thereby limiting topological element changes during the watermarking process.

The frequency-domain watermarking algorithm uses invariant geometric information in key frequency-domain coefficients to embed watermarks. Common frequency-domain algorithms include the discrete Fourier transform, discrete wavelet transform and discrete cosine transform. The discrete Fourier transform based watermarking algorithm for vector maps is to embed watermarks in the Fourier coefficient sequence, which take advantage of the Fourier description factor's geometric invariance on the scaling and rotation to promote the watermarking scheme's robustness [27–29]. Kitamura et al. [30] divided vector maps into grid sets as analogue to the pixels in raster images and introduced discrete wavelet transform to embed watermark in vector maps. Yang and Zhu [31] decomposed map coordinates based on discrete wavelet decomposition and then embedded watermarks into low-frequency coefficients with the retrieval of watermarked vector data using inverse wavelet transformation. Voigt et al. [32] proposed a reversible watermarking algorithm for vector maps based on the discrete cosine transform. As there are high correlations among vertices belonging to the same polygon in a vector map, the algorithm performs the discrete cosine transform on individual data block composed of every eight vertices and embeds watermark in the cosine change coefficients.

Vector map watermarking algorithms based on both the coordinate domain and the frequency domain directly operate the coordinate points. Even if changes to the original map data are controlled within given ranges, errors are inevitably introduced for subsequent analysis and thus the overall accuracies of the vector maps are reduced. In addition, to control the map accuracies, the watermark capacities of both the coordinate-domain and frequency-domain algorithms are relatively small.

Reversible watermarking and zero-watermarking developed in recent years are two popular watermarking algorithms that have zero-disturbance characteristics. Reversible watermarking of vector maps is an extension of the coordinate domain algorithm. The watermark message is embedded in the least significant bit by the difference between two adjacent points. When the copyright information is verified, the watermark is extracted according to the rule of judging the difference between two points and the original data are then restored after extraction [33]. The reversible watermarking technique embeds the watermark information in all vertices and therefore it allows for increasing the watermark capacity, and as a result, scholars have developed various large-capacity reversible watermarking schemes [34–36]. Because reversible watermarking completely removes watermark information from the embedded data after data restoration, meaning that the watermarking can be used only once, the approach does not satisfy the user demand for permanent watermarking. Zero-watermarking

could overcome the drawback by constructing watermarks based on feature information of digital data and storing watermarking information in the authoritative third-party protection center [37]. Because zero-watermarking does not directly embed watermarks in digital data, it does not alter the original data and thus possesses the advantages of zero-disturbance. Zero-watermarking has already been applied to various data types [38–40] and offers opportunities for the applications in vector maps with high fidelity. There are pioneer studies on zero-watermarking of vector maps and most of them mainly focus on the mining and construction of stable feature information [41–43]. The zero-watermarking of a vector map often has the problems of small watermark capacity and lack of robustness because of complex types of attacks to vector maps and the limitation posed by the stability and distribution of feature information. The idea of multiple-watermarking [44,45] that integrates multiple algorithms to embed watermarks based on diverse requirements in different embedding domains [46] can theoretically overcome these shortcomings. Compared to the individual watermarking scheme, multiple-watermarking is more functional and robust to attacks as well as increasing the watermark capacity. Note that multiple-watermarking typically alters the original data greatly and multiple-watermarking based on the zero-watermarking scheme could perfectly avoid influencing the original data. Designing a dual zero-watermarking scheme for vector maps with large capacity and multi-attack resistance could offer an effective watermarking method to meet the application needs.

The objectives of this study are to: (1) construct a dual zero-watermarking scheme that synthesizes complementary information from both feature and non-feature vertices of 2D vector maps; (2) assess the proposed scheme by performing various attack experiments to the watermarked vector maps.

## 2. Methodology

The processing steps include the preprocessing of the vector maps, encryption of watermark information, construction of zero-watermarks based on both feature and non-feature vertices, and watermark extraction and assessment. Data preprocessing primarily includes map blocking and feature extraction. Watermark encryption is to scramble watermark images and the serialization of the encrypted information. The construction of two types of single zero-watermarking schemes is mainly based on the mining of the feature information of the feature vertices and non-feature vertices. Finally, watermark extraction and inspection methods mainly introduce extraction steps and related test indicators.

### 2.1. Data Preprocessing

Steps for preprocessing of the original vector map primarily include map blocking and feature vertex extraction. Map blocking refers to dividing the map into blocks for subsequent uses of embedding watermarks into each individual block, which is helpful to defend against map cropping attacks. Feature vertex extraction is to extract stable vertices that are key elements in vector maps for subsequent uses of watermark construction [42,43].

The used method of map blocking directly impacts on the distribution and capacity of the watermarks. Different algorithms of map blocking have been developed, primarily including uniform blocking (UNIF), quadtree blocking (QUAD), and modified quadtree block (MQUAD) [1]. UNIF divides the vector map into several rectangles with equal grid areas. QUAD and MQUAD divide the vector map adaptively by quarters to force the vertex number in each block to be greater than prescribed thresholds. Considering the complexity and inhomogeneity in the spatial distribution of vertices, this study applies the vertex-constrained blocking method [47], a UNIF-based method, to split vector maps into blocks. The method continues to divide the maps into blocks until the vertex number within individual blocks are less than a given threshold. The initial block area $S$ and the threshold of vertex number $P$ are important to map blocking. If the values of $S$ and $P$ are set too large, the number of watermarks in the data will be reduced, and if the values of $S$ and $P$ are set too small, the blocking efficiency is low and the blocks are too scattered. Compared with typical UNIF or QUAD methods,

the vertex-constrained blocking method generates reasonably distributed blocks that contain similar vertex numbers as well as evenly distributed watermarks in subsequent analysis.

Feature vertices of vector maps are key elements that contain the major feature information of the map and are not removable under attacks. The less vulnerable the selected feature points, the stronger the robustness of the corresponding watermark. Extraction of feature vertices could be viewed as the process to remove non-feature vertices. In this study, the Douglas-Peucker algorithm, a typical algorithm for simplifying line features, is used to extract feature vertices. Line simplification is also the basic method of vector map compression, the greater the threshold set by the Douglas-Peucker algorithm, the stronger the anti-compression attack ability of the watermarking scheme, but the fewer feature vertices obtained, the less the watermark capacity. Therefore, before zero-watermark is constructed, an appropriate distance threshold value needs to be selected through trial and error to ensure that there are sufficient and stable watermark information in each block. On this basis, the non-feature vertices can be obtained by erasing the feature vertices of the original vector map.

*2.2. Watermark Encryption*

To prevent the watermark information from being easily identified and attacked, the initial watermark image W needs encryption. The Arnold algorithm [48] is a classic scrambling algorithm that encrypts the watermark image, which transforms the coordinates (x,y) of each vertex on a N × N watermark image into (x′,y′) as follows:

$$
\begin{bmatrix} x' \\ y' \end{bmatrix} = \begin{bmatrix} 1 & b \\ a & ab+1 \end{bmatrix} \begin{bmatrix} x \\ y \end{bmatrix} mod(N) \tag{1}
$$

where *a* and *b* are positive integers, mod is the remainder function, *N* is the width of the matrix. These parameters determine the scrambling cycles.

This study chooses a binary image with copyright information as the initial watermark image *W* and scrambles it to obtain an encrypted watermark image *W′* based on the Arnold algorithm. The encrypted watermark is transformed into an ordered sequence that consists of only 0 and 1 and is further divided into r groups with 8-bit digits in each group. The 8-bit digits in each group are transformed into an 8-bit unsigned integer I (ranging from 0 to 255). The 8-bit unsigned integers of the *r* groups together form a watermark sequence $W_n$, which stores the encrypted watermark information and is used to generate zero-watermarks in subsequent operations.

*2.3. Zero-Watermark Based on Delaunay Triangle Mesh*

Given that topological relation and angles in vector maps are resistant to geometric attacks [49,50], this study first constructs Delaunay Triangulation Mesh (DTM) based on the feature vertices and then produces zero-watermark of high capacity and strong robustness by using angles in the DTM as feature information the feature vertices [51]. The specific steps are as follows:

The DTM is first generated from the feature vertices of each block and the distances between adjacent vertices are derived based on (2). Angles of the triangles in DTM in blocks are then obtained using inverse cosine functions (3) to form an angle sequence $A_n$ (4), of which the length is greater than that of the watermark sequence $W_n$ in each block (Figure 1).

$$
\begin{cases}
D_{12} = \sqrt{\left(p_1^x - p_2^x\right)^2 + \left(p_1^y - p_2^y\right)^2} \\
D_{23} = \sqrt{\left(p_2^x - p_3^x\right)^2 + \left(p_2^y - p_3^y\right)^2} \\
D_{31} = \sqrt{\left(p_3^x - p_1^x\right)^2 + \left(p_3^y - p_1^y\right)^2}
\end{cases} \tag{2}
$$

$$\begin{cases} \theta_1 = arc\cos\left[(D_{12}^2 + D_{31}^2 - D_{23}^2)\,/\,(2\cdot D_{12}\cdot D_{31})\right] \\ \theta_2 = arc\cos\left[(D_{23}^2 + D_{12}^2 - D_{31}^2)\,/\,(2\cdot D_{23}\cdot D_{12})\right] \\ \theta_3 = arc\cos\left[(D_{12}^2 + D_{31}^2 - D_{23}^2)\,/\,(2\cdot D_{12}\cdot D_{31})\right] \end{cases} \tag{3}$$

$$A_n = \{\,\theta_1,\ \theta_2,\ \theta_3\cdots,\ \theta_k\,\} \tag{4}$$

where $p_1^x$, $p_2^x$, $p_3^x$ are the X-axis values of the first, second and third vertex, $p_1^y, p_2^y, p_3^y$ are the Y-axis values of the first, second and third vertex, $D_{12}, D_{31}, D_{23}$ are distances between two different vertex in a same triangle, $\theta_1, \theta_2, \theta_3$ are three angle values of the same triangle and $A_n$ is a set of angle values of each block.

If the length of $A_n$ is several times longer than the watermark sequence $W_n$, $d$ watermark sequences would be supplemented at the end of $W_n$ to extend the watermark length. For example, if the length of $A_n$ is 3300 and the length of $W_n$ is 800, three message bit string of $W_n$ would be extended after $W_n$, and the watermark length becomes 3200. Applying the operator of exclusive disjunction (XOR) to $A_n$ and $W_n$ (5) would then generate zero-watermarking images $Z_n$ (i.e., the watermark secret key) for the corresponding blocks. Both the original watermarking images and zero-watermarking images are then saved to an authoritative third-party agency IPR (e.g., the Intellectual Property Rights center) with added time stamps of legal copyrights to defend against interpretation attacks [52].

$$Z_n = A_n \oplus W_n,\ n = 1,\ 2,\ 3, \cdots, d \tag{5}$$

When detecting watermarks, it only requires applying the same compression method to extract feature vertices for constructing DTM such that the same watermarks can be obtained by applying the same operations. As long as the feature vertices are unchanged, the constructed DTM is unique for a vector map no matter where to start constructing the DTM of the vector map. The characteristic of DTM greatly reduces the complexity of constructing triangle networks and avoids losing synchronicity when experiencing geometric attacks. Because attacks such as addition or deletion to feature vertices would only affect adjacent triangles and have no impacts on other triangles, the use of DTM could maintain local stability well.

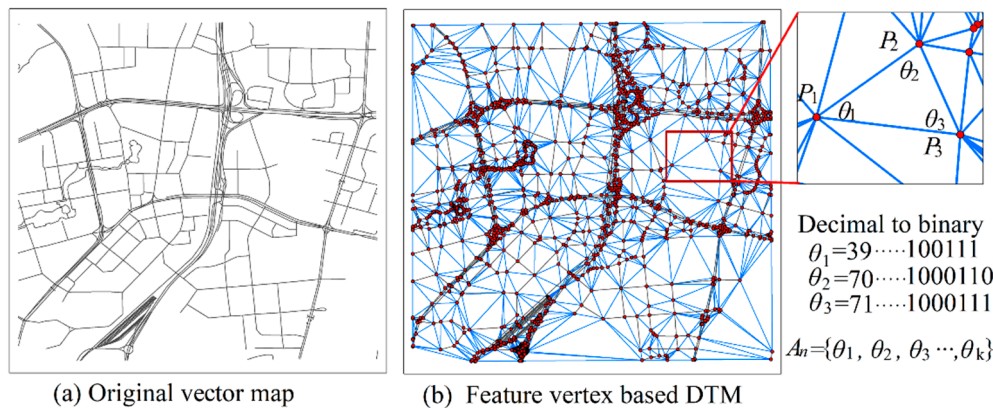

(a) Original vector map　　　　(b) Feature vertex based DTM

**Figure 1.** Construction of Delaunay triangulation mesh based on feature vertices. DTM: Delaunay Triangulation Mesh.

### 2.4. Zero-Watermark Based on Singular Value Decomposition

Singular Value Decomposition (SVD) is an orthogonal transformation of matrix without requirements for the matrix size. Watermarking algorithms based on SVD are robust against perturbation as SVD transformation produces matrix that are stable to the operation of transpose, displacement and rotation [53]. The singular values reflect intrinsic map features that represent relations among matrix elements and therefore zero-watermark constructed based on non-feature vertices and SVD is robust to common geometric attacks.

Taking the horizontal direction as an example, where the operation to the vertical direction is analogous, steps to implement SVD to construct zero-watermark are follows:

Based on non-feature vertices in all polylines and polygons, a fixed-size real matrix $C = (C_{i,j}) \in R^{M \times N}$ is constructed for each non-feature vertex set P of every polyline and polygon. The matrix of C is decomposed by applying the SVD algorithm (6).

$$C = U \begin{bmatrix} \Sigma & 0 \\ 0 & 0 \end{bmatrix} V^T = [u_1, u_2, u_3 \cdots, u_k] \begin{bmatrix} \sigma_1 & 0 & & 0 \\ & \sigma_2 & & \\ & & \ddots & \\ & & & \sigma_k \\ 0 & & & 0 \end{bmatrix} [v_1, v_2, v_3 \cdots, v_k]^T \quad (6)$$

where $[u_1, u_2, u_3 \cdots, u_k]$ and $[v_1, v_2, v_3 \cdots, v_k]$ denote their left and right eigenvectors, respectively; the diagonal matrix $\Sigma = diag(\sigma_1, \sigma_2 \cdots \sigma_k)$ satisfies $\sigma_1 \geq \sigma_2 \geq \cdots \geq \sigma_r \geq \sigma_{r+1} \cdots \geq \sigma_k = 0$; r denotes the rank of $\Sigma$ and equals to the number of non-zero singular values; $\sigma_i$ is a unique singular value of the matrix $C$ as determined by decomposition.

Because the first singular value $\sigma_1$ contains the most information of the matrix elements, meaning that it is more robust than other singular values, the first singular value $\sigma_1$ of the matrix C, let $S = \{\sigma_1^f\}$, $f = 1, 2, \cdots, m$ (m is the number of polylines and polygons, S is the set of the first singular values from all polylines);

Let $\overline{\sigma_1} = \frac{1}{m}\sigma_1^f$, the mean of all the first singular values $\overline{\sigma_1}$ can be obtained, and subsequently compare the size of $\sigma_1^f$ and $\overline{\sigma_1}$ to get a binary table $B_f$, as follows:

$$B_f = \begin{cases} 1, & \sigma_1^f > \overline{\sigma_1} \\ 0, & \text{otherwise} \end{cases} (f = 1, 2, \cdots, m) \quad (7)$$

Through the XOR operation between the binary table $B_f$ and the binary sequence of $W_n$, a zero-watermark $Z_f$, which is consistent with the size of the watermark image, is obtained. If the number of the first singular values is less than the number of watermark bits, the first singular values would be reused orderly to construct zero-watermark. Similarly, a legally validated timestamp is added to the zero-watermark $Z_f$ and is saved to the IPR center.

$$Z_f = B_f \oplus \text{Bit}(W_n), f = 1, 2, \cdots, m. \quad (8)$$

*2.5. Watermark Detection*

Watermark detection for copyright protection involves watermark extraction, the inverse process of watermark construction, and watermark inspection, the comparisons between watermark images extracted from the vector maps and watermark images stored in the copyright center.

To extract watermarks, vector maps to be detected are preprocessed into individual blocks and both feature and non-feature vertices are obtained. Zero-watermarks $Z_n$ and $Z_f$ are obtained from the Intellectual Property Rights center and the grouping and transformation processes are performed to obtain the watermark secret keys $Z'_n$ and $Z'_f$, respectively. Following the same processes as watermark construction, both DTM and SVD are processed to extract the angle sequence $A'_n$ and the singular value binary table $B'_f$. The operators of exclusive disjunction (XOR) are performed between the watermark sequence $Z'_n$ and the angle sequence $A'_n$, and between the watermark sequence $Z'_f$ and the singular value binary table $B'_f$, respectively, such that encrypted watermark images are obtained according to the watermark sizes. The watermark image $W'$ for the vector maps to be detected can then be obtained from the encrypted watermark images after inverse operation of the Arnold scrambling.

Watermark inspection needs to make comparisons between watermarks extracted and stored, whereas main indictors in this study used for assessment include watermark similarity, bit error rate

and watermark capacity. The Normalized Correlation (*NC*) coefficients are used to assess the similarity between images [11] as follows:

$$NC = \frac{\sum_{i,j} W_{i,j} * W'_{i,j}}{\sqrt{\sum_{i,j} W^2_{i,j}} \sqrt{\sum_{i,j} W'^2_{i,j}}} \tag{9}$$

where $W_{i,j}$ and $W'_{i,j}$ are stored and extracted watermarks at the coordinates of $(i, j)$, respectively.

The Bit Error Ratio (*BER*) is a quantitative metric to evaluate the specific error size of extracted watermark [54]:

$$BER = \frac{100}{L} \sum_{i,j=0}^{L-1} \begin{cases} 0 & W_{i,j} = W'_{i,j} \\ 1 & W_{i,j} \neq W'_{i,j} \end{cases} \tag{10}$$

where *L* is the bit length of the watermark information.

The metrics of *NC* and *BER* are useful to assess the accuracy of the watermarks. *NC* ranges from 0 to 1, where high *NC* denotes strong similarity. When the vector maps are not altered, *NC* is 1 and *BER* is 0. If being attacked by different operations, the higher the *NC* is and/or the smaller the *BER* is, the easier the watermark can be recognized, and vice versa.

Watermark capacity is also an important aspect to assess the ability of a watermarking scheme on protecting digital data. Given that vertex are the basic elements for vector maps, an indicator to watermark capacity is defined as the ratio between number of embeddable bits and number of vertices.

## 3. Experimental Design

### 3.1. Experimental Data

This experiment uses a binary image with a copyright character as the original watermark information. The image size is 80 × 80 pixels and the length of the watermark sequence $W_n$ is 800. The parameters of both a and b in (1) are set as 1 and the original watermark image was implemented with traversed scrambling. Figure 2 shows the watermark images after 10, 30, 45 and 60 times of Arnold scrambling. After 60 times of Arnold scrambling, the transformed image was restored to the original watermark image. The 10 times scrambling of the watermark image was chosen as the encrypted watermark (Figure 2b) for the use of constructing and assessing the dual zero-watermarking for the 2D vector maps.

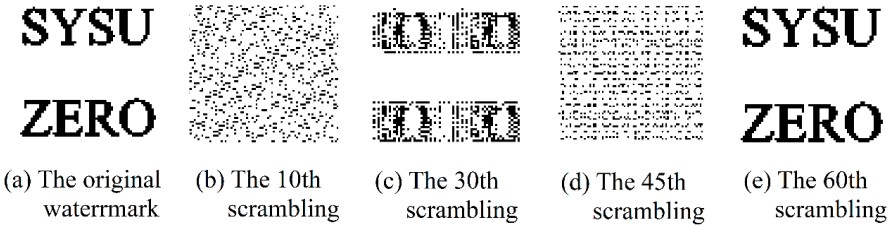

| (a) The original wattermark | (b) The 10th scrambling | (c) The 30th scrambling | (d) The 45th scrambling | (e) The 60th scrambling |

**Figure 2.** Arnold scrambling of the watermark image.

Two different vector maps were used to evaluate the proposed scheme. One 2D vector map of the water distribution in Zengcheng (Figure 3, denoted by ZC) was used to represent environments with irregular and uneven distributed map elements, and another 2D vector map of the road distribution in Shenzhen (Figure 3, denoted by SZ) was used to represent environments with regular and relatively evenly distributed elements. The projections of the vector maps are local plane projection coordinate systems, namely Zengcheng independent coordinate system for ZC and Shenzhen independent coordinate system for SZ, respectively. Given that polygons could be seen as polyline sets, we obtained 77,343 vertices and 1848 polylines in ZC, 128,069 vertices and 4683 polylines in SZ.

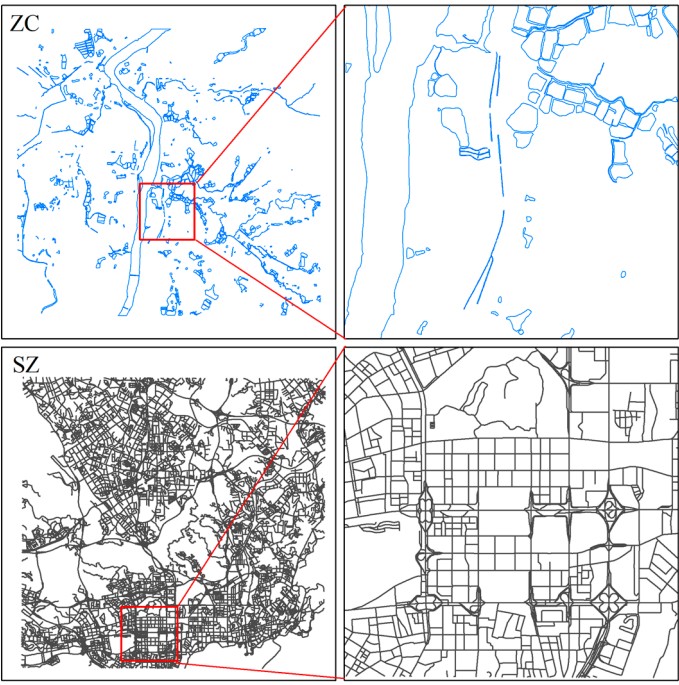

**Figure 3.** Original 2D vector maps.

## 3.2. Watermark Construction

First, we implement the vertex constraint blocking process on ZC and SZ. The initial sub-block area is 1/4 of the total map, and the number of vertices threshold P is 5000. To enhance the resistance to compression attack, the Douglas-Pecuker algorithm chooses a large threshold, and the compression ratios are 77.99% and 78.98%, respectively. By constructing the DTM and operating XOR, we could obtain 26 blocks and 42 zero-watermarks in ZC, and 38 blocks and 68 zero-watermarks in SZ. Based on implementing the Douglas-Pecuker algorithm, all non-feature vertices are reversely extracted, and all polylines that contain non-feature vertices are traversed to construct the SVD matrix. Considering the number of non-feature vertices on all polylines, a SVD matrix with the size of $4 \times 5$ is constructed for each polyline in both ZC and SZ to obtain the zero-watermarks according to the construction steps (Table 1). This method makes full use of the non-feature vertices that are often ignored in the conventional zero-watermarking schemes and does not influence zero-watermarks based on feature vertices. As zero-watermarks generated based on conventional methods distribute regionally, this method that uses almost all polylines could provide zero-watermarks that are distributed across the entire map, thereby helping the overall copyright protection.

**Table 1.** Information of the original maps and the constructed watermarks.

| ID | Vertex | Feature Vertex | Polylines | Blocks | $Z_n$ | $Z_f$ |
|----|--------|----------------|-----------|--------|-------|-------|
| ZC | 77,343 | 17,020 | 1848 | 26 | 42 | 1 |
| SZ | 128,069 | 26,913 | 4683 | 38 | 68 | 1 |

## 3.3. Attack Experiment Design

Different from other digital data, the 2D vector maps could face various operational attacks in applications, including both changes in displayed graphics and relevant attributes. The attack experiments here therefore include a variety of attacking operations such as geometric attack, vertex attack, object attack and compound attacks. In the subsequent analysis, the watermark with the highest similarity or the lowest bit error rate is selected in the attack experiment for the performance evaluation of the watermarking scheme.

3.3.1. Geometric Attack Experiment

Geometric attacks mainly include translation, rotation, scaling, coordinate transformation, compression, and cropping attacks. The attacking intensities and modes are shown in Table 2 and Figure 4 In the coordinate transformation attack, both ZC and SZ datasets are converted into the Xian80 and Beijing54 coordinate systems, of which both are commonly used in China, respectively. Four levels of compression attack experiments, of which 78% is the threshold for constructing the current scheme. For cropping attacks, several cropping modes as described in [24] are applied to facilitate comparisons. This study designed six cropping patterns for attacking experiments (Figure 4) and extracted watermark information from the remaining vector data. Both NC and BER values are calculated to assess the robustness against attacks.

**Table 2.** Magnitudes of attacks. CST: Coordinate System Transformation.

| Attack Types | Range of Magnitude |
| --- | --- |
| Translation | 0–100 on x-axis or y-axis |
| Rotation | $0°-360°$ |
| Scaling | $-500\%-500\%$ |
| CST | Transform to Beijing54 and Xian80 |
| Compression | Compression ratios are 30%, 50%, 78% and 88% |
| Cropping | Multiple cropping patterns show in Figure 4 |

**Figure 4.** Different cropping patterns. Pattern 1 cropped the lower half of the original maps by approximately 1/2. Pattern 2 cropped the right half of the original maps by approximately 1/2. Pattern 3 left approximately 1/4 of the upper left corner of the original maps. Pattern 4 left approximately 1/4 of the first half of the original maps. Pattern 5 left approximately 1/4 of the right half of the original maps. Pattern 6 left approximately 1/8 of the upper right of the original maps.

### 3.3.2. Vertex Attack Experiment

Vertex attack is a common operation to vector maps that could result in coordinate changes, including interpolation, simplification, and, registration. Different from the method that remove non-feature vertices by the Douglas-Pecuker algorithm, this experiment adds and delete vertices randomly to result in random influence on both feature and non-feature vertices. The experiment intensity ranges from 5% to 60% of the original maps. The same intensity attacks for both addition and deletion were performed randomly for three times, where the DTM-based feature vertex watermarks, the SVD-based non-feature vertex watermarks, and the combined watermark were extracted accordingly.

### 3.3.3. Object Attack Experiment

The object attack experiment mainly includes object deletion, object addition, and reordering of polyline vertices. The vector map watermarking scheme needs to resist object attacks such as deletion or addition of polylines, which are routine operations in cartography and map updating. The object addition experiment is to add polylines randomly to ZC and SZ at the ratio ranging from 5% to 60%. Different Object IDs are assigned to the added polylines to make them have certain attribute meanings and participate in relevant batch processing. In the object deletion experiment, polylines in ZC and SZ are randomly deleted in proportion, and the deletion intensity is increased gradually at the range from 5% to 60%. Similar to the vertex attack, both object deletion and addition experiments are performed for three times randomly with the same intensity, whereas the feature vertex watermarks, the non-feature vertex watermarks, and the dual watermarks are extracted accordingly. The reordering of objects and/or vertices is one type of attack that does not result in changes in images or data precision, but easily causes false traversal orders and influences the more common objects in vector geographic data. In Figure 5, two types of vertices reordering are performed, for both ZC and SZ.

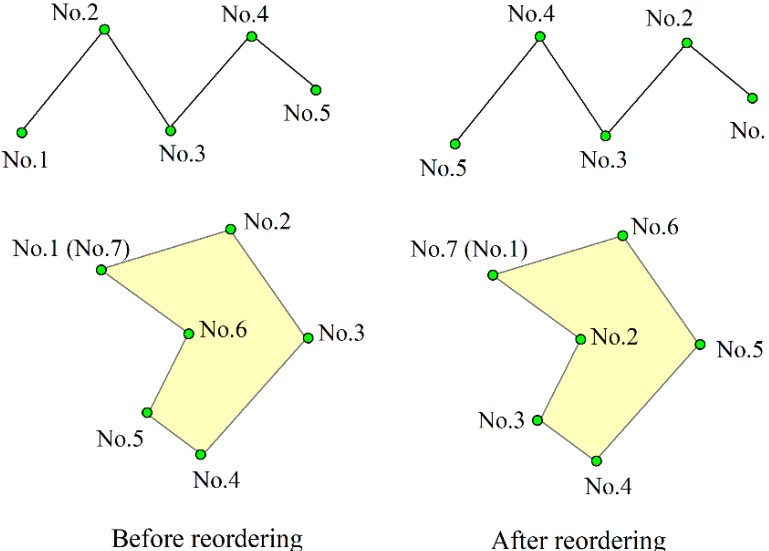

**Figure 5.** Object reordering.

### 3.3.4. Compound Attack Experiment

In practical applications such as urban planning, map updating, and visual analysis, 2D vector maps may suffer from a wide range of attacks. In the compound attack experiment, 2–8 types of attack methods are randomly performed for both ZC and SZ, where the attack methods include the above-mentioned geometric attack, vertex attack, and object attack with random attacking intensities. Similarly, to assess the proposed scheme, each attack is simulated randomly for three times to extract the feature vertex zero-watermark, the non-feature vertex zero-watermark and the dual zero-watermark.

## 4. Results and Discussion

### 4.1. Watermark Capacity

Table 3 compares the watermark capacity for different schemes, of which four schemes designed for large-capacity are selected from literature for comparisons. The method in Peng et al. [34] conforms to the basic capacity of the reversible watermark, which reaches 1 bit/vertex. The methods in both [35] and [36] add the embedding algorithms, which increase the average capacity to 1.97187 bit/vertex and 3.9978 bit/vertex, respectively. The zero-watermarking scheme developed in [55] could reach an average watermark capacity of 1 bit/vertex, nearly as high as the reversible watermarking. The watermark capacities in the above-mentioned schemes are at the high-end of methods in the literature. By comparison, the watermark capacity of our developed scheme could reach 3.50316 Bit/vertex on average, which is approximately 3.5 times that in [34,55], 1.8 times that of Wang's method, and only slightly lower than that of Lin's method.

**Table 3.** Watermark capacity of different methods (bit/vertex).

| Data Set | 2D Vector Data Set | Capacity | Basic Algorithm | Reference |
|---|---|---|---|---|
| G1 | Engineering Graphic | 1 | reversible watermarking | Peng et al. (2011) [34] |
| G2 | Engineering Graphic | 1 | | |
| G3 | Engineering Graphic | 1 | | |
| Average | Engineering Graphic | 1 | | |
| Coastline map | Geospatial data | 1.98271 | reversible watermarking | Wang et al. (2014) [35] |
| Road map | Geospatial data | 1.97605 | | |
| Windmill Islands map | Geospatial data | 1.95686 | | |
| Average | Geospatial data | 1.97187 | | |
| G1 | Engineering Graphic | 3.9981 | reversible watermarking | Lin et al. (2018) [36] |
| G2 | Engineering Graphic | 3.9986 | | |
| G3 | Engineering Graphic | 3.9956 | | |
| Average | Engineering Graphic | 3.9978 | | |
| An engineering graphic | Engineering graphic | 1 | zero-watermarking | Du and Peng (2008) [55] |
| ZC | Geospatial data | 3.55818 | zero-watermarking | The proposed |
| SZ | Geospatial data | 3.44814 | | |
| Average | Geospatial data | 3.50316 | | |

The watermark capacity directly affects the watermark robustness. In general, the larger the watermark capacity, the lower the robustness, but in our repeat embedding scheme, more capacity can promote a higher chance of the complete watermark information retained under attacks. For the coordinate-domain and frequency-domain watermarking algorithms, the more information the embedded watermark contains, the easier the map accuracies could change. As such, the watermark capacities for both the coordinate-domain and frequency-domain watermarking algorithms are elastic and typically low. The watermark capacity is however an important metric for assessing the non-interference watermarking schemes such as reversible watermarking and zero-watermarking, which are compared in Table 3. The design and processing of our proposed scheme help improve the watermark capacity, for example, the vertex-constrained blocking method makes a dense uniform watermark distribution and improves the number of embedded watermarks, and the construction of DTM creates a large amount of feature information that could provide a large space for constructing the zero-watermarks. In addition, the design of dual zero-watermarking increases the watermark capacity as compared to the single watermarking scheme. All these aspects that reflect the developed scheme could enhance the watermark capacity.

## 4.2. Watermark Robustness

### 4.2.1. Resistance to Geometric Attacks

The experimental results for both ZC and SZ in Table 4 show that under different levels of translation, rotation, and scaling attacks, both DTM-based and SVD-based schemes could detect the watermark images. Under the coordinate transformation attacks, the *NC* values for the optimal feature vertex-based watermark are 0.84 and 0.83 in ZC, respectively, and 0.85 and 0.84 in SZ, respectively, when transforming to the Xian80 and Beijing54 coordinate projection. Although the corresponding images are relatively vague, we can still identify the copyright information. The *NC* values of the non-feature vertex watermarks obtained from these two maps are 1 under the coordinate transformation attack. Under the compression attacks, the DTM-based watermarking scheme can extract a 100% similar watermark image at a compression ratio of 78%. The watermark quality decreases as the compression strength exceeds 78%. In the case of a 50% compression rate, the extracted SVD-based watermark can hardly recognize the copyright information, and when the compression rate reaches to 78%, it is impossible to construct non-feature vertex zero-watermark.

**Table 4.** Results of geometric attacks.

| Attack | NC(DTM)/(a) | $W'$ | NC(DTM)/(b) | $W'$ | NC(SVD)/(a) | $W'$ | NC(SVD)/(b) | $W'$ |
|---|---|---|---|---|---|---|---|---|
| Translation 0–100 | 1 | SYSU ZERO | 1 | SYSU ZERO | 1 | SYSU ZERO | 1 | SYSU ZERO |
| Rotation 0°–360° | 1 | SYSU ZERO | 1 | SYSU ZERO | 1 | SYSU ZERO | 1 | SYSU ZERO |
| Scaling −500%–500% | 1 | SYSU ZERO | 1 | SYSU ZERO | 1 | SYSU ZERO | 1 | SYSU ZERO |
| Transfer coordinate to Xian80 | 0.84 | SYSU ZERO | 0.85 | SYSU ZERO | 1 | SYSU ZERO | 1 | SYSU ZERO |
| Transfer coordinate to Beijing54 | 0.83 | SYSU ZERO | 0.84 | SYSU ZERO | 1 | SYSU ZERO | 1 | SYSU ZERO |
| Compression 30% ratio | 1 | SYSU ZERO | 1 | SYSU ZERO | 0.94 | SYSU ZERO | 0.95 | SYSU ZERO |
| Compression 50% ratio | 1 | SYSU ZERO | 1 | SYSU ZERO | 0.71 | SYSU ZERO | 0.73 | SYSU ZERO |
| Compression 78% ratio | 1 | SYSU ZERO | 1 | SYSU ZERO | - | - | - | - |
| Compression 88% ratio | 0.89 | SYSU ZERO | 0.82 | SYSU ZERO | - | - | - | - |

DTM is constructed based on the stability of the topological relationship among feature vertices such as that the angle values in the DTM remain the same as the vector map does not distort under operations of translation, rotation, or scaling. Singular values of non-feature vertices could change, but the relative size remains unchanged. The projective transformation has a high intensity of vertex operations, but the *NC* values for the SVD-based watermarks are almost unaffected as the relative sizes of the vertices' values have little change. The *NC* values of the DTM-based watermarks however are affected greatly, where the extracted watermark images have the *NC* values ranging between 0.83 and 0.85. The copyright information can still be identified and the watermarks in ZC are affected

slightly more than that in SZ as the original projection coordinate system and the degrees of projection transformation differ. The number of feature vertices is not affected by changes in the compression strength when the compression strength does not exceed a threshold and decreases as the compression strength increases after exceeding the threshold. As non-feature vertices could not be extracted, the watermarks for non-feature vertices could not be constructed. In essence, watermarks based on DTM exhibit robust to the compression attacks.

Table 5 shows the *BER* values of the optimal watermarks that can be detected in ZC and SZ in the proposed methods and in the methods from literatures of [1] and [24] under six cropping patterns, where there are results for two different embedding methods in [24]. Under the six cropping patterns, the watermark based on DTM could be used to extract the watermark image with a *BER* value of 0, meaning that the extracted watermark information is completely consistent with original watermark information, which indicates a strong resistance capability to the cropping attacks. Correspondingly, the *BER* values of the watermarks extracted from the developed dual zero-watermarking scheme is 0 under different cropping attacks. The non-feature vertex watermarks have very high *BER* values under different cropping attacks, indicating a low resistance capability to the cropping attacks. The *BER* values obtained under the method in literature [1] in the corresponding patterns are notably better than that for the non-feature vertex watermark. The methods in [24] provide *BER* values lower than those obtained from the method in [1] for all cropping attacks except both Pattern 1 and Pattern 4. Overall, watermarks constructed based on the proposed DTM method or the dual methods are far superior to the comparative methods in literatures in terms of resistance to the cropping attack.

**Table 5.** Results of cropping attack experiment.

| Cropping | ZC/DTM | ZC/SVD | SZ/DTM | SZ/SVD | The Proposed | Ohbuchi et al. (2002) [1] | Ohbuchi et al. (2003) [24] | |
|---|---|---|---|---|---|---|---|---|
| **Pattern 1** | 0 | 45.0 | 0 | 43.5 | 0 | 1.4 | 0.0 | 0.0 |
| Pattern 2 | 0 | 43.6 | 0 | 44.6 | 0 | 1.8 | 0.1 | 0.4 |
| Pattern 3 | 0 | 62.6 | 0 | 65.0 | 0 | 7.0 | 0.7 | 1.3 |
| Pattern 4 | 0 | 67.0 | 0 | 63.6 | 0 | 6.8 | 0.4 | 0.0 |
| Pattern 5 | 0 | 67.3 | 0 | 63.9 | 0 | 11.3 | 1.4 | 0.4 |
| Pattern 6 | 0 | 76.1 | 0 | 75.5 | 0 | 35.9 | 18.8 | 15.1 |

The DTM-based feature vertex watermarks have a large capacity and are uniformly distributed, in these cropping patterns, at least one complete sub-block is included, therefore, the complete watermark information can be extracted. Non-feature vertex watermarking relies on Polylines where all non-feature vertices are located, under these six cropping patterns, the remaining parts are less than 50% of the original maps. Because the content is missing too much, it is not enough to extract an effective non-feature vertex watermark, so the *BER* values are also very high. This scheme combines the advantages of feature vertex watermarking and non-feature vertex watermarking, therefore, the complete watermark with a *BER* of 0 can be extracted in the six cropping patterns. The vertex-constrained blocking in the data preprocessing section contains at least one complete watermark information, the remaining blocks under these six cropping patterns are much larger than the smallest block under preprocessing. Therefore, the ability of this scheme to resist cropping attacks is more than this experiment in theory.

### 4.2.2. Resistance to Vertex Attack

The *NC* values for both non-feature vertex zero-watermarks of ZC and SZ are 1 under different levels of vertex addition attacks (Figure 6a), and the corresponding trend lines for ZC/SVD and SZ/SVD remain straight at *NC* = 1. The feature vertex zero-watermarks have *NC* values greater than 0.8 for attacks under the intensity of the first 40% and the first 50% for ZC and SZ, respectively. The SZ/DTM curve is higher than the ZC/DTM curve. The watermark qualities gradually reduce with increased attack intensity. The *NC* values for the dual zero-watermarking scheme are always 1 under

different levels of vertex addition attacks, which are the same as the trend of the SVD-based watermark. As seen from Figure 6b, the feature vertex watermark, the non-feature vertex watermark and the dual zero-watermark all show to be robust under the vertex deletion attack. The watermarks based on feature vertex show a slow downward trend for both the ZC and SZ datasets and have the *NC* values greater than 0.8 for the deletion intensity up to 55%. The watermarks based on the non-feature vertex perform better than those based on the feature vertex given that their *NC* values keep above 0.8 with a slower declining trend than those of ZC/DTM and SZ/DTM. The dual zero-watermarks combine the strengths of watermarks constructed based on both feature and non-feature vertex and have the same trends as the SVD-based watermark that shows robust to vertex deletion.

In the process of extraction, we find that when implementing feature vertex extraction by the Douglas-Peucker algorithm, the random increased vertices cannot be compressed, but as the feature vertices to build DTM, so vertex addition attack has no effect on the non-feature vertex watermark. The feature vertex watermark based on the abundant feature information and local stability, early insertion of fewer vertices had little effect on it. When the increasing strength of the attack affects the stability of all local watermarks, the NC values of feature vertex watermark decreased greatly. The feature vertex watermark in SZ is better than it is in ZC, which benefits from the larger feature information in SZ to increase the uncertainty of vertex addition attack, so the local watermarks in SZ are superior. In the proposed scheme, the number of feature vertices is much fewer than non-feature vertices, which determines the randomly deleted feature vertices are very few, so the DTM-based feature vertex watermark shows good robustness against vertex deletion attacks. Reliant on the stability of singular values and the characteristic of scheme, the non-feature vertex watermark also shows good robustness under vertex deletion, even obviously better than the feature vertex watermark. The optimal watermark extracted from the dual zero-watermark is almost the same as the non-feature vertex watermark, so it inherits the robustness advantage of the non-feature vertex watermark in the vertex attack.

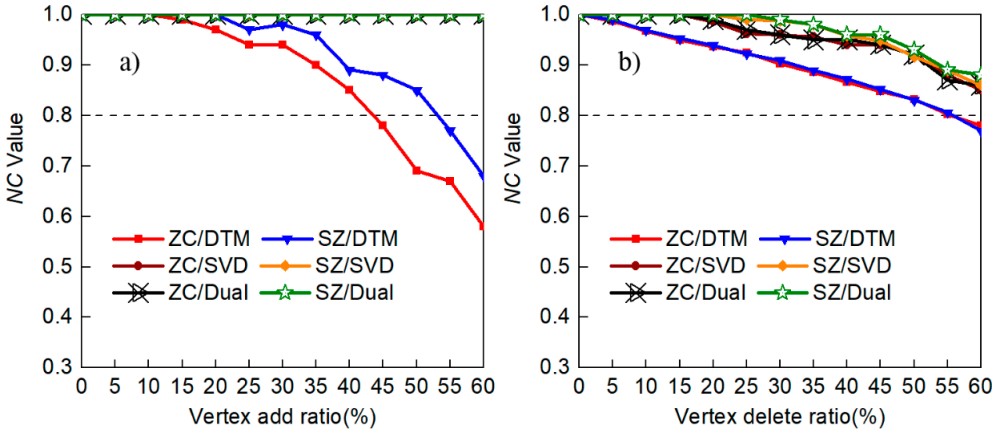

**Figure 6.** The *NC* change curves under increased levels of (**a**) randomly added vertices and (**b**) randomly deleted vertices.

### 4.2.3. Resistance to Object Attack

Under reordering attacks, the *NC* values for the DTM-based watermarks keep as 1 and those for the SVD-based watermarks are all higher than 0.99, implying that the proposed scheme is effective against reordering attacks. The proposed watermarking scheme does not include object nor vertex identification, as such, the reordering attack has no effect on the feature sequence or the generated watermark. Because the SVD matrix used in this study has a fixed matrix size, reordering attacks could have influence on the singular value of the SVD matrix if an object contains more vertices than the matrix.

As shown in Figure 7a, our proposed method performs well under the object addition attack for both ZC and SZ. The *NC* values of the feature vertex watermarks, namely ZC/DTM and SZ/DTM, keep above 0.84 within the attack range and only decline slowly as the attack intensity increases. The *NC* values of the non-feature vertex watermarks, namely ZC/SVD and SZ/SVD, are always 1. Consistent with the non-feature vertex watermarks, the dual zero-watermarking schemes always have the *NC* values of 1 under the object addition attack. In Figure 7b, the DTM-based watermarks have the *NC* values of 1 for the first 35% deletion rate of the feature vertex watermarks and the quality of the watermarks greatly reduces as object deletion enhances. When more than 45% of objects are removed, watermarks could not be detected efficiently as the *NC* values are lower than 0.8. The SVD-based watermarks could not be effectively detected when the deletion ratio exceeds 25%. The dual zero-watermarking scheme maintains the same trend as the feature vertex watermark, showing that the proposed scheme is robust against both object addition and deletion.

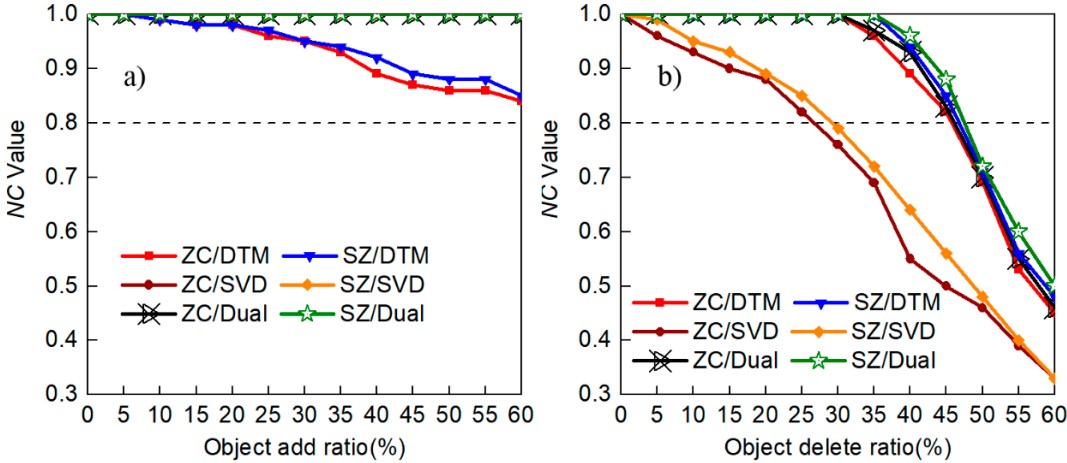

**Figure 7.** The *NC* change curves under increased levels of (**a**) object addition and (**b**) object deletion.

### 4.2.4. Resistance to Compound Attack

Figure 8 shows the distribution of the *NC* values for three types of watermarks in ZC and SZ under multiple compound attacks. The overall distribution trends under random compound attacks as shown in Figure 8a and b are similar, showing a downward trend with increased attack numbers. The *NC* values for both the DTM-based zero-watermarks and dual zero-watermarks roughly distribute between 0.5 and 1, whereas the SVD-based zero-watermarks have a range from 0 to 1. The *NC* values for the dual zero- watermarking scheme are significantly higher than those of both the feature vertex watermark and the non-feature vertex. Figure 9 shows that the ratio distributions of the watermarked images have *NC* values greater than 0.8 under compound attacks. As the attack numbers increase, the probability for detecting high-quality watermark images decrease and the chances to detect valid watermark information, even with eight attacks, is still non-negligible.

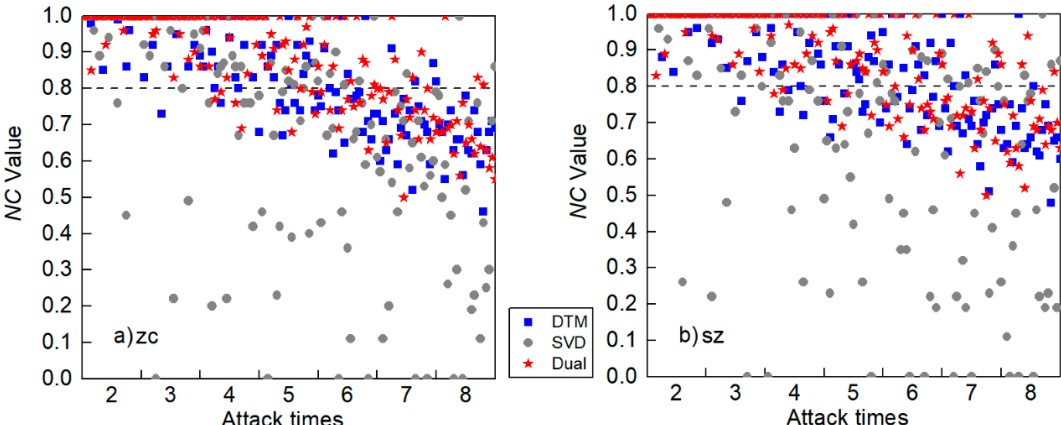

**Figure 8.** The scattered distributions of the *NC* values for the feature vertex watermarks, the non-feature vertex watermarks, and the dual watermarks under compound attacks in (**a**) ZC and (**b**) SZ.

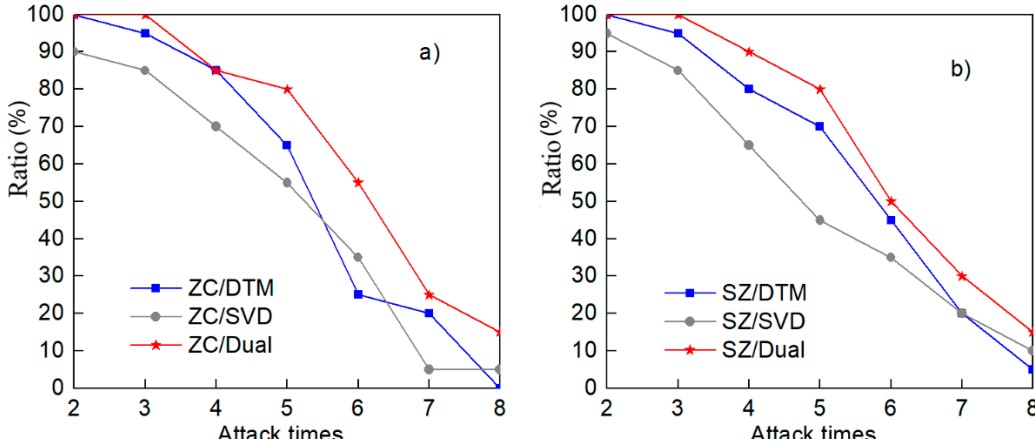

**Figure 9.** The ratio distribution of watermarks for *NC* higher than 0.8 under different numbers of compound attack in (**a**) ZC and (**b**) SZ.

### 4.3. Robustness Complementarity

This study not only designed two zero-watermarking schemes that are not conflicting and show strong robustness against certain attacks, but also show complementary robustness. The dual zero-watermarking scheme combines the advantages of the two individual zero-watermarking schemes and improves the robustness as reflected in the compound attack experiment, including geometric attacks, vertex attacks and object attacks, where the dual zero-watermarking scheme performs better than the two independent schemes. For example, in a geometric attack, the coordinate transformation attack does not impact on the non-feature vertex watermark such as that the *NC* values remain to be 1. By comparison, the coordinate transformation attack could influence the feature vertex watermark greatly as the *NC* values distribute between 0.83 and 0.86. For the compression attack, the feature vertex watermarks could be well detected but the non-feature vertex watermarks could not be extracted if the compression strength exceeds a certain threshold of watermark construction. For both vertex addition and deletion attacks, the non-feature vertex watermarks perform better than the feature vertex watermarks. The non-feature vertex watermark performs better than the feature vertex watermark under object addition attacks, but worse under object deletion attacks. In all these cases, the feature vertex and non-feature vertex watermarking schemes show to be independent and complement, reflecting the superiority of the dual zero-watermarking scheme.

*4.4. Uniqueness Test*

The zero-watermarking algorithm does not directly embed watermark information into the original data but relies on the feature information in the original maps to construct zero-watermarks, as such, the generated zero-watermarks must be unique to ensure the uniqueness of copyright verification. To test the uniqueness of zero-watermarks under different algorithms and in different datasets, the zero-watermarks generated by watermarking schemes in ZC and SZ are extracted and used to verify each other. As shown in Table 6, the extracted watermark information is compared and the obtained *NC* values are between 0.37 and 0.41 with no valid watermark information identified, meaning that zero-watermarks generated using different datasets and algorithms are identical.

**Table 6.** Cross check results (*NC* values).

| Watermarked Data | | $W_n^{'}$(ZC/DTM) | $W_n^{'}$(ZC/SVD) | $W_n^{'}$(SZ/DTM) | $W_n^{'}$(SZ/SVD) |
|---|---|---|---|---|---|
| Zero-watermarks | $Z_n$(ZC/DTM) | 0.4008 | 0.3936 | 0.3805 | 0.3794 |
| | $Z_n$(ZC/SVD) | 0.4039 | 0.4005 | 0.3905 | 0.3881 |
| | $Z_f$(SZ/DTM) | 0.4009 | 0.3884 | 0.3872 | 0.3986 |
| | $Z_f$(SZ/SVD) | 0.3886 | 0.3895 | 0.3923 | 0.3863 |

## 5. Conclusions

This study designs a dual zero-watermarking scheme of 2D vector maps. The scheme uses both feature vertices and non-feature vertices to construct zero-watermarks and improves the watermark capacity. The scheme combines the topological feature information among the feature vertices and the SVD eigenvalues derived from the non-feature vertices, and perform well in geometric attacks, vertex attacks, and object attacks. The overall robustness of the proposed dual zero-watermarking scheme is better than individual schemes as the two individual schemes are complementary in terms of robustness. The scheme shows to be robust for protecting the 2D vector maps as tested.

**Author Contributions:** The authors contributed equally to the paper.

**Funding:** This research was supported by the National Natural Science Foundation of China (grant nos. 41671453), the National Natural Science Foundation of China (grant nos. 41431178), the National Natural Science Foundation of China (grant nos. 41875122), the Natural Science Foundation of Guangdong Province in China (grant no. 2016A030311016) and the National Key R&D Program of China (grant no. 2017YFA0604302), the Research Institute of Henan Spatio-Temporal Big Data Industrial Technology (grant no. 2017DJA001).

**Acknowledgments:** We thank all reviewers for their constructive comments.

**Conflicts of Interest:** The authors declare no conflict of interest.

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
