# Peer review of "Dual Zero-Watermarking Scheme for Two-Dimensional Vector Map Based on Delaunay Triangle Mesh and Singular Value Decomposition"

_applsci, doi:10.3390/app9040642_

Round 1
Reviewer 1 Report
The manuscript presents a dual zero-watermarking scheme for vector maps. In the proposed method, firstly, the feature vertices are extracted and a zero-watermarking algorithm based on Delaunay Triangulation Mesh (DTM) is applied. Secondly, for non-feature vectors, a singular value decomposition (SVD)-based zero-watermarking scheme is used. The combined strategy is shown to obtain remarkable capacity and robustness results in quite a convincing way, and my overall evaluation of the manuscript is, thus, positive. However, there are a few issues that I would like the authors to address in a revised version of this work.
My major comments about the manuscript are the following:
1. The scheme is reported to embed several copies of the watermark in block-wise fashion. Hence, the zero-watermark ($Z_n$) occurs several times in the image (at least once per block), and sometimes more than once in each block (if my interpretation of the data in Table 1 is correct). The other zero-watermark $Z_f$ is obtained for non-feature vertices. I assume the $Z_f$ can also be quite long and, thus, several instances of the image watermark may be contained in $Z_f$. When retrieving the watermark, only one value of NC and BER is given for $Z_n$ and $Z_f$. The method somehow combines the different instances of the extracted watermark image into a single image. No detail is provided in the manuscript about this process. Is there any kind of voting scheme? Some discussion is needed.
2. Regarding the experiments, there is a nice comparison between the capacity of the proposed method (Table 3) with other schemes. However, when robustness is discussed (in Section 4.1), no general comparison with other schemes (either reversible or zero-watermarking) is given. Only a comparison is provided in Table 5 wrt to two rather old methods, and only taking into account cropping attacks. In order to improve the manuscript, the robustness results of the proposed scheme should be compared with those of recent state-of-the-art methods. This may imply to obtain the implementation of some of the schemes (or to implement them).
3. Section 4.2.4 discusses the resistance of the method agains a composite attack. According to the manuscript, Fig. 9 represents the "probability of surviving a sequence of attacks". First, I think that "probability" is not the right word to use here, since the authors are probably referring to a ratio of survival and, thus, the analysis is empirical and not theoretical. "Ratio" would be a better concept to use here. Secondly, the authors mention that the chance to detect the watermark is still high even after 8 attacks. According to Fig. 9, the detection "probability" after 8 attacks is in the range 10-20% and, thus, I would not say it still high. It may be expressed as "non-negligible".
4. Regarding the cropping attacks reported in Table 5, the BER values are increasing from 43.5% (pattern 1) to 76.1% (pattern 6). It is, however, unexpected to have values much greater than 50%. In the worst case (random guess), on average, half of the bits would be extracted correctly, whereas the other half would be flipped. This would lead to BER equal to 50%. Actually, a value of 76% is as good as one of 24%, since the inverse of the extracted bits could be used instead of the extracted values. I would like to know whether the results in Table 5 are correct and, if so, what is the reason for the results much above 50%.
5. On page 11, the authors claim that "The larger the watermark capacity, the higher chance the complete watermark information is retained under attacks." This is typically NOT true. In general, the higher the capacity, the lower the robustness. However, if you use repeat embedding and a voting scheme, a higher robustness may be obtained with more capacity. This should be clarified in the manuscript.
6. On page 12, we find the following sentence: "The SVD-based watermark decreases with increased compression level and the watermark cannot be constructed when the compression level exceeds 78%." In fact, from Table 4, the SVD-based watermark seems to be destroyed even for a compression ratio of 78% (not higher than 78%). We only know that the watermark survives for compression ratios up to 50%.
As minor corrections, please revise (rephrase or rewrite) the following sentences and phrases:
a) Page 2:
- "There however lacks of consideration for attacks under element deformation."
- "The watermark byte is embedded": maybe "message" instead of "byte"
b) Page 3:
- "If $S$ and $P$ are set too large, the number of watermarks would be reduced correspondingly, and if $S$ and $P$ are set too small, the algorithm efficiency is low and scattered high."
c) Page 4:
- "divided into r groups": please use mathematical formatting for $r$.
d) Page 7:
- "The pixel size is 80 x 80" --> "The image size is 80 x 80 pixels"
e) Page 11:
- "in literatures" --> "in the literature"
- "in average" --> "on average"
Author Response
Thank you for all your comments. All the mistakes you mentioned about the sentence, word or format have been corrected, and the loose part of the rule description has been deleted or rewritten. You mentioned emphatically that there is no discussion on the reasons for some results in the paper, we have added some contents to strengthen the interpretation of the results and the algorithm. Please refer to the Response and the revised manuscript for details.

Reviewer 2 Report
1. Even though the overall level of English is understandable, there are several places where it can be improved. On top of this, there are several typos through the whole text
Line 39, the word 'imperceptibility' is better than 'invisibility'
Line 56, a full-stop is missing
Sentence from line 65 to line 68 is not clear
Line 117, I think it should be "construction of zero-watermarks.."
2. The Methodology section needs more details, otherwise it is not possible to reproduce the results using the same two vector-map images used in this paper
how are feature vertices selected (how are parameters adjusted) with the Douglas-Peucker algorithm in each block?
In section 2.2, why is the watermark converted to 8bit uint when it is already binary? It is used for XOR operation which is binary. Even though your implementation might use conversion to 8bit, it is not necessary for the XOR-ing later
In section 2.3, how are the DTM triangles (and the 3 angles in them) ordered in the final sequence An? The order is important to be unique so that it can be reproduced during detection.
In line 190, you mention "applying the same compression method". Does the detector need to have information about the specific level of compression in each block? (previously it was mentioned that the Douglas-Peucker is adjusted in each block)
In Equation (5), does n should go from 1 to d, or from 1 to d+1, instead from 1 to 3?
Typo in the caption of Figure 1a
In section 2.4, how is the matrix C constructed? What are the values in it? Again, how do you order the singular values in Bf (or how do you order the polygons and polylines) - should it be unique so that it can be reproduced during detection?
Lines 240 and 244, it is 'bit error rate'
In Equation (11), what is bit? It should be mentioned clearly. Or it can be written descriptively without using an equation, e.g. "capacity is defined as the ratio between number of embeddable bits and number of vertices"...
3. Section 4 needs more discussions in few places
there are two sections 4.1
why the DTM/SVD watermark is perfectly robust to rotation/translation? The angles are certainly not changed during rotation/translation, but how is the order of extracted feature vertices preserved? Or how is the block division preserved? Does the detector use additional information to restore the order?
Cropping may preserve a full DTM block and ensure perfect detection, but how the synchronization during detection is achieved from a cropped image? I suppose the division into blocks (of the cropped image) is different from the non-cropped one. Does the detector use additional information to restore the original blocks before cropping?
Line 428, "the..zero-watermarks.. are 1" - this should be changed. For example, "the NC values for both zero-watermarks are 1..."
Why the SVD watermark is resistant to vertices addition? Needs to be commented why. Added vertices may be non-feature vertices (as said in line 318), and impact on the synchronization/order of extracted SVD features. The same goes for object addition attack - why the SVD watermark is perfectly robust?
Line 477, "less than six times" is just a concrete example, not something that applies generally. Either needs to be specified that it is only for the ZC image, or be removed. I would prefer removing it, because even for the ZC image, for 7 attacks the probability of DTM is again higher than SVD
Section 4.4, the uniqueness test is weak, for example, it could include high number (around 100-1000) of randomly generated Wn to try to authenticate the given 2 vector-map images
Author Response
Thank you for all your comments. We have accepted and corrected all the errors in statements, words, formats and other aspects as required by your advices. In the comments, you mentioned for many times the preservation order of vertex, feature information, watermark information and the retrieval mode after the attack. Based on the preservation mode of GIS vector data and the data processing principle of arcpy, we made a specific explanation and added explanatory statements in the paper as required. Please refer to the Response and the revised manuscript for details.

This manuscript is a resubmission of an earlier submission. The following is a list of the peer review reports and author responses from that submission.